# Development of a Nakazima Test Suitable for Determining the Formability of Ultra-Thin Copper Sheets

**Nejia Ayachi** [1,2]**, Noamen Guermazi** [2]**, Cong Hanh Pham** [1] **and Pierre-Yves Manach** [1,*]

[1] UMR CNRS 6027, IRDL, Université Bretagne Sud, Rue de Saint Maudé, F-56100 Lorient, France; nejia.ayachi@univ-ubs.fr (N.A.); conghanhckp@yahoo.com (C.H.P.)

[2] LGME, ENI Sfax, Route Soukra Km 3.5 BP 1173, 3038 Sfax, Tunisia; noamen.guermazi@enis.tn

**\*** Correspondence: pierre-yves.manach@univ-ubs.fr

**Abstract:** The objective is to propose an accurate method for determining the forming limit curves (FLC) for ultra-thin metal sheets which are complex to obtain with conventional techniques. Nakazima tests are carried out to generate the FLCs of a pure copper and a copper beryllium alloy with a thickness of 0.1 mm. Because of the very small thickness of the sheets, the standard devices and the know-how of this test are no longer valid. Consequently, new tools have been designed in order to limit friction effect. Two different methods are used and compared to estimate the necking: the position-dependent measurement method (ISO Standard 12004-2), and the time-dependent method based on the analysis of the derivatives of the planar strain field. It is shown that the ISO standard method underestimates the forming limit curves. As the results present non linear strain paths, a compensation method is applied to correct the FLCs for the tested materials, which combines the effects of curvature, nonlinear strain paths and pressure. The curvature effect for such thickness and punch diameter on the FLCs is weak. The results show that this procedure enables to obtain FLCs that are close to those determined by the reference Marciniak method, leading to a minimum in major strain that converges to the plane strain state.

**Keywords:** ultra-thin sheets; Nakazima tests; forming limit curves; copper alloys

---

## 1. Introduction

The considerable increase in even smaller and lighter products is pushing several industrial sectors to move toward the field of miniaturization [1]. Forming by plastic deformation of metal parts with very small dimensions using ultra-thin sheets is playing an increasingly important role in the connectivity and electronics industry in particular [2]. The thickness of a sheet is what determines whether it is called thick, thin or ultra-thin. For sheet suppliers, a sheet is ultra-thin if its thickness varies between 0.013 mm and 0.2 mm. In the scientific literature, this definition depends not only on the thickness but also on the average grain size of the material, or rather the number of grains to thickness ratio [3]. The use of ultra-thin sheets in the manufacture of small parts is limited by several factors such as material, tools, friction, etc. The issues observed in micro-forming are strongly associated with miniaturization, since in their forming processes, some intrinsic dimensions of material such as the grain size and the surface roughness are not scaled down with the geometric dimensions [4].

When modelling a forming process by plastic deformation, it is relevant to know the limit strains that a given material can support [5]. For example, knowledge of the limits of formability is crucial [6] because it allows it to be used to the best of its ability without ever exceeding these [7]. The forming limit diagram (FLD) is one of the most useful tools to evaluate the formability of sheet

metals [8,9] as well as the location of the points representing principal strains with respect to the forming limit curve [10]. FLCs, which represent these limits, therefore is a classical diagnostic tool for the study of a metal sheet [11]. The formability is referred to as the ability to resist localized necking [12]. This concept is based on the use of the two curves, that of Keeler and Backofen (in the field of expansion) [13] and that of Goodwin (in the field of restraint) [14], representing the forming limit curve of Keeler-Goodwin [15]. In the space of the main strain tensor [16], one can separate the successful zone, located below the curve, and the zone of failure located above the FLC, where the formed parts present defects [2].

FLCs are usually determined from tests on specimens of different shapes to cover the entire field of deformations in the sheet plane [17]. Various experimental set-ups have been developed to reproduce different loading paths by varying the geometry of the tools and that of the specimens [18]. The most widely used are the Marciniak test [19], which uses a cylindrical punch with a counter-blank having a circular hole, positioned between the punch and the blank, in order to promote breaking in the central (flat) part of the sheet, and the Nakazima test [20] which uses a hemispherical punch [21] and blanks of different widths held between a die and a circular blank holder. The advantages of this last test are the simplicity of the tools and blanks, which makes it possible to obtain all the loading paths necessary to build a FLC [22]. The experiments carried out by Grolleau et al. [23] using V-bending, notched tension and Nakazima specimens on 2024-T351 aluminum specimens, DP450 steel and DP980 steel show that for these materials, the Nakazima test provides a reliable estimation of the strain at fracture for the plane stress state. None of the two other tests (V-bending and notched tension) provides reliable results for the three materials. As a result, only with the help of the Marciniak and Nakazima tests, the forming limit curves can be fully drawn, despite that in the case of the Nakazima test there are sometimes difficulties related to the measurement of the deformations. However, the curvature of the blanks after forming, due to the hemispherical shape of the punch, and the friction between the blank and the tools can skew the deformation measurements in the critical zone [24]. For each sheet, the necking limits can be influenced by the strain rate and the temperature, given the sensitivity of these two parameters of the material [25]. Moreover, recently many studies have also noted that process-dependent parameters, such as non-linear strain paths, sheet curvatures, contact pressure, etc., can affect necking limits [26]. For instance, Abspoel et al. [27] stated that the strain paths in Nakazima tests are not exactly proportional, which affects limit strain measurement in comparison with results of Marciniak testing. Min et al. [24] showed that the differences observed in limit strains obtained from Nakazima and Marciniak tests are caused by a combination of effects from non-linear strain paths, sheet curvature, and in the case of the Nakazima test, contact pressure between the punch and sheet prior to the onset of localized necking.

The effect of the strain paths on the position of the FLC is thus of high importance. The role of this parameter is essential in sheet metal forming because the manufacture of complex parts frequently requires several passes, which induces non-linear strain paths. Many studies have pointed out that non-linear strain paths affect the forming limit strains, starting with the foundational work of Nakazima [28]. The latter has proposed a solution to this problem, by a method of accounting for the strain path effect based on the work-equivalent plastic strain. Others [29–31] proposed to evaluate formability of the sheet based on stress states rather than strain states [28]. The effect of curvature has been noticed in the first applications of the FLC [10]. Min et al. [24] evaluated the severity of the forming conditions on each inner, middle and outer layer to determine the conditions giving rise to localized necking instability in a curved sheet. For this, they determined the forming limit of the Nakazima test, which has a high curvature, for all layers of the thickness in order to determine the least critical layer. They showed that the stress in the middle layer of the thickness was the appropriate strain metric to use in comparaison to the strain FLD. Another process parameter that influences the determination of the FLC is pressure. It is generally accepted that the pressure across the thickness influences localized necking, but the identification of this influence is no longer obvious because of the often observed conflation between localized necking and fracture [24]. In fact, the presence of

a negative stress across the sheet thickness, caused by the contact pressure between the sheet and the Nakazima punch, will delay the onset of necking. The latter can generate greater plane strain in order to obtain the condition of localized necking instability. By accounting for the effects of these processing conditions, Min et al. [24] were able to derive a single forming limit curve for both Marciniak and Nakazima tests for conditions of perfectly linear strain paths restricted to in-plane stretching under plane-stress conditions.

In this paper, Nakazima tests are performed on specific samples with a designed-on-purpose device; a reduced scale is used for all tools and samples to adapt to ultra-thin sheets, in order to accurately determine the FLC of a pure copper and a copper beryllium alloy with a thickness of 0.1 mm. Indeed, because of the very small thickness of the sheets, the classical dimensions of the tools are irrelevant, and the standard devices and the know-how of this test are no longer valid. Consequently, new tools have been designed in order to limit friction effect and machined to perform these tests. The forming limit strains are measured through the major and minor strains associated with the onset of necking along a given strain path. Two different methods are used to estimate the necking: the first one is the position-dependent measurement method, which is proposed by the ISO Standard 12004-2 [32], and the second one called time-dependent, based on the analysis of the derivatives of the planar strain field. The FLC points resulting from the ISO 12004-2 method are compared with those obtained by the time-dependent method. Due to the smallest curvature of the punch compared to classical tools, the results present non-linear strain paths as well as other process-dependent effects. A compensation method is thus applied, which combines the effect of curvature, non-linear strain paths and pressure. According to the procedure described in Min et al. [24], a numerical subroutine was developed to correct the FLCs for the tested materials. The new FLC obtained after compensation of the process-dependent effects are analyzed in the light of works already published on thicker steel sheets [24,33].

## 2. Experimental Procedure

### 2.1. Materials

The materials tested in this study are a pure copper (Cu 99.9%) and a copper beryllium alloy (CuBe2). According to Goodfellow supplier, the copper is pure at 99.9%, and the sheets are cold rolled to a thickness of 0.1 mm. Quantitative information on the microstructure is obtained from Electon BackScatter Diffraction (EBSD) scans in the rolling and transverse direction sections. The device used in this study is a 7001 Field Electron Gun Scanning Electron Microscope from Jeol (JEOL Ltd., Tokyo, Japan) equiped with an Oxford EBSD CCD camera. EBSD data were post-treated with the software Channel 5 from Oxford. The scanned areas were $320 \times 240 \ \mu m^2$ large and the step size was set to 0.3 μm. It is observed that the texture is weakly marked, as shown in Figure 1. The average grain size is 10 μm with a maximum of 40 μm. The dispersion on the grain size is low with an average of 10 grains in the thickness.

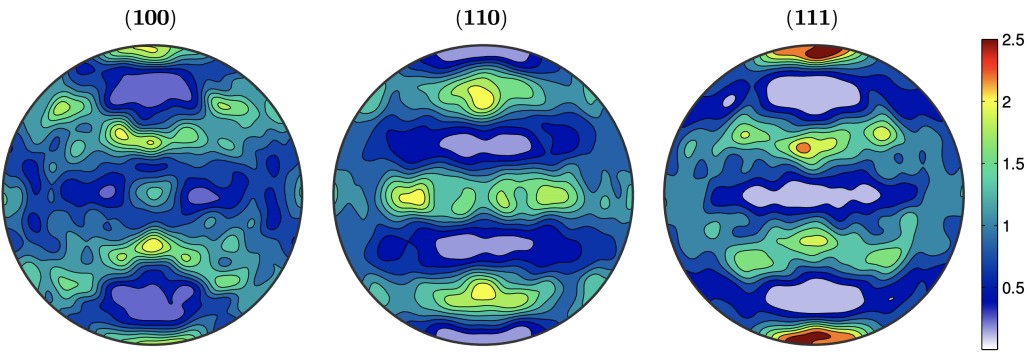

**Figure 1.** Pole figure of the pure Cu 99.9% material. Directions are indicated above the figures.

The other material is the copper beryllium alloy CuBe2 which chemical composition is given in Table 1. The cold rolled sheet is 0.1 mm thick which is not annealed after cold rolling. Quantitative information on the microstructure was also obtained from EBSD scans. The microstructure was measured in the rolling-transverse and transverse-normal sections. No sensible microstructure gradient was observed and the grains were considered equiaxed. A pole figure of the microstructure is shown in Figure 2. The experimentally measured average grain size was about 4 μm. Although the maximum grain size was about 50 μm, little dispersion was observed and the grain size was quite homogeneous leading to an average of 25 grains throughout the sheet thickness.

**Table 1.** Chemical composition in mass percent of CuBe2.

| Cu | Be | Co | Ni | Fe |
|----|------|-----|------|------|
| 97 | 1.8–2 | 0.3 | 0.15 | 0.15 |

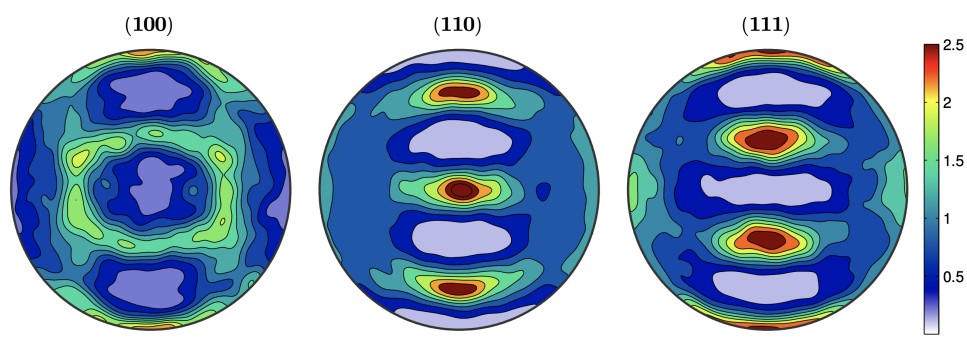

**Figure 2.** Pole figure of the copper beryllium alloy CuBe2. Directions are indicated above the figures.

The hardening behaviour has been characterized in tension for several orientations to the rolling direction. A sample geometry was prepared according to ISO 6892-1 standard, similarly to [34]. Three samples were tested and the results were averaged in each direction. Strains were measured using Digital Image Correlation (DIC) system Aramis 4M (GOM GmbH, Braunschweig, Germany). The usual strain-hardening parameters corresponding to Hollomon's Equation (9) were obtained by fitting the uniaxial Cauchy stress-Logarithmic plastic strain tensile curves. The results are presented for both materials in Figure 3.

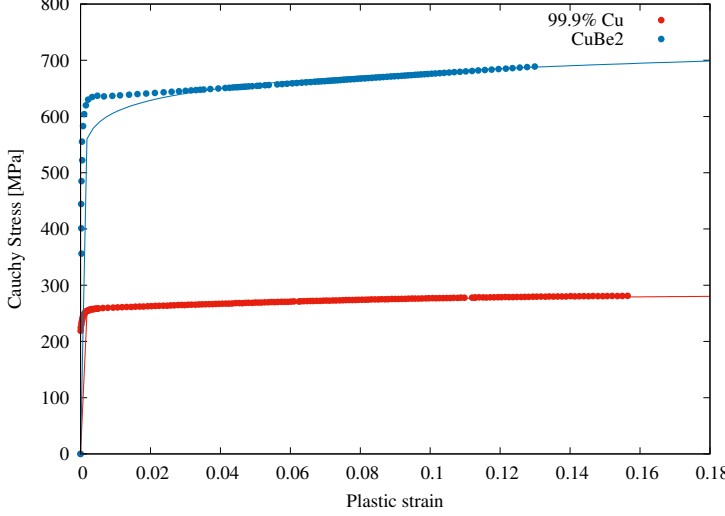

**Figure 3.** Hardening curves in a tensile test in the rolling direction and the identified Hollomon's model for pure Cu and CuBe2 alloy.

The plastic anisotropy coefficients (r-values) were determined from simultaneous measurement of longitudinal and transverse strains of tensile samples having different orientations to the rolling direction, and the assumption of volume invariance. The mechanical properties of the pure Cu and CuBe2 alloy are listed in Table 2. All experimental tests were conducted at room temperature.

**Table 2.** Mechanical properties of the sheet materials. YS: Yield Stress, UTS: Ultimate Tensile Stress, $\bar{r} = (r_0 + 2r_{45} + r_{90})/4$, $\Delta r = (r_0 - 2r_{45} + r_{90})/2$ and $K, n$ are the Hollomon's parameters.

| Material | YS (MPa) | UTS (MPa) | $\bar{r}$ | $\Delta r$ | $K$ (MPa) | $n$ |
|----------|----------|-----------|-----------|------------|-----------|-----|
| Cu 99.9% | 247 | 285 | 0.69 | 0.08 | 340 | 0.11 |
| CuBe2 | 430 | 590 | 0.92 | 0.28 | 787 | 0.18 |

## 2.2. Nakazima Tests

Nakazima tests have been performed in this work. These tests consist in deforming sheet metal blanks of different geometries using a hemispherical punch until fracture occurs. Because of the very small thickness of the sheets, the classical dimensions of the tools (punch diameter of 100 mm) have become irrelevant and the devices and the know-how of this test which apply for sheets from 0.4 to 3 mm thick, are no longer valid. Consequently, a new device has been designed in order to limit friction effects and machined for these tests as described in [3]. One main advantage of this test compared to Nakazima or Marciniak devices that have been developed specifically for such ultra-thin sheets [35] is that it remains macroscopic, avoiding any local or scale effects that can arise with a very small punch. A schematic drawing of the Nakazima testing device is depicted in Figure 4a with dimensions of the tools and the specimens reduced by a scale of 1/3 compared to those defined in the ISO 12004-2 Standard [32]. During the deformation process, the samples were fixed by a constant blank-holder force (23 kN), while the punch was moved against the blank at a velocity of 0.25 mm·s$^{-1}$ to deform the specimen until its fracture.

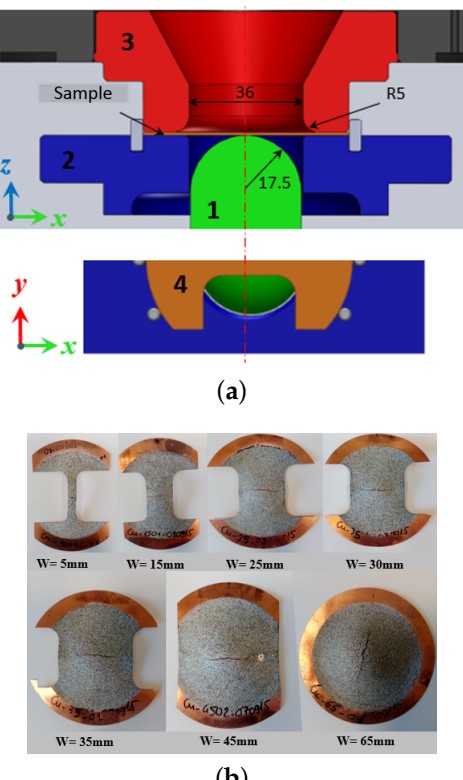

**Figure 4.** (**a**) Left-hand part: tools of the Nakazima test, (1) punch, (2) blank-holder, (3) die, (4) sample. (**b**) Right-hand part: geometries of the samples for the Nakazima tests.

By varying the specimen width, different forming conditions occur on the sheet surface, from a regular biaxial deformation to a simple tensile strain state. The effective width (*w*) of the samples varies from 5 to 65 mm (Figure 4b). Each test configuration was performed at least three times to check its reproducibility.

A parasitic phenomenon of Nakazima tests is the friction between the punch and the blank, that may cause the rupture outside the center of the specimen. This issue can prevent its measurement by the digital image correlation (DIC) measurement system. In order to reduce this phenomenon, a Teflon film with a thickness of 0.05 mm was bonded to the lower surface of the sample via a layer $MOS_2$ grease. Another layer of this grease was also deposited on the punch.

The strain distributions are obtained by using the DIC system ARAMIS 60 Hz developed by GOM. An area of $4 \times 4$ mm$^2$ in the middle of the sample is selected to determine both major and minor strains. For each specimen configuration, the strain path is followed by plotting the evolution of major and minor strains along a section aligned with the longitudinal direction of the sample, up to fracture. The results are plotted for each material in Figures 5 and 6.

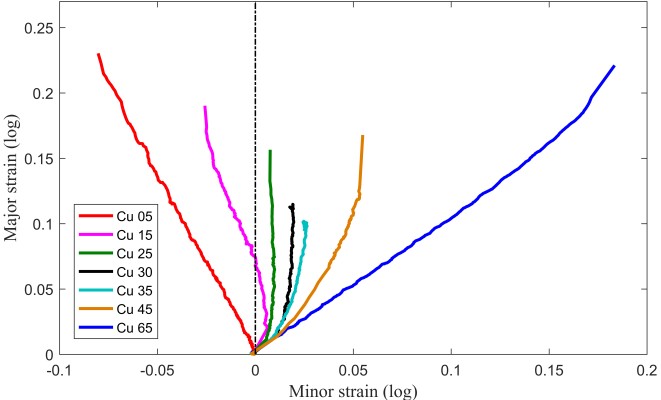

**Figure 5.** Evolution of major-minor strains of pure copper specimens from 5 mm width (Cu 05) to 65 mm (Cu 65) in the Nakazima tests.

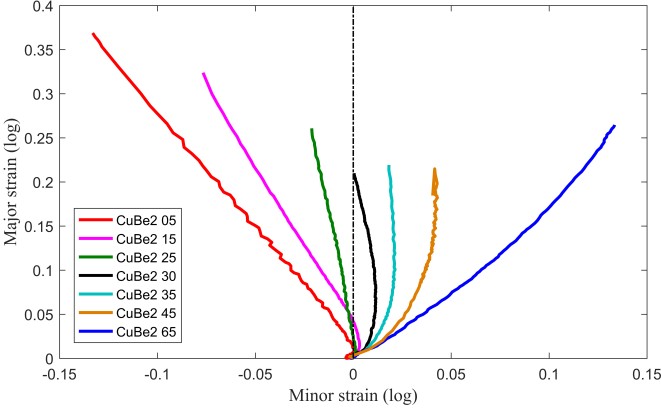

**Figure 6.** Evolution of major-minor strains of CuBe2 alloy specimens from 5 mm width (CuBe2 05) to 65 mm (CuBe2 65) in the Nakazima tests.

It can be observed that due to the geometry of the punch, most of the strain paths are non-linear for both materials. Indeed, limit major strains associated with the onset of localized necking obtained from the Nakazima tests are higher than those from the Marciniak test [24]. A smaller punch decreases friction effects in the Nakazima test but results in higher limit major strains. Inconsistent limit strains from different forming limit test methods are attributed to the effects of sheet curvature, non-linear strain path, and contact pressure in the area of the neck in the Nakazima tests. It is observed that the non-linearity of the strain paths is more sensitive for pure copper, which supposes larger friction

effects for this material. This result is consistent with its lower number of grains in the thickness (10 to 25 for CuBe2) which leads during plastic deformation to an increase of the surface roughness [36].

### 2.3. Experimental Determination of Forming Limit Curves

#### 2.3.1. ISO 12004-2 Method

The ISO 12004-2 Standard [32] provides a method for determining FLCs from the strain state of the specimen just before its rupture. This method is also referred to as the position-dependent measurement method [7]. Three sections are defined perpendicular to the break line to calculate the FLC data, with a distance equal to 1.5 mm between each two successive sections according to the procedure defined in [3]. From these section lines the major and minor strain values are obtained for the evaluation of the corresponding FLC data points.

Using the major strain $\epsilon_1$ and minor strain $\epsilon_2$ section data sets, an inverse parabola is fitted through two fit windows on both sides where the crack appears, as presented in Figure 7. The maximum values of the polynomial functions are the values of the strain at necking. The main advantage of this method is that it is only based on the analysis of strain fields. One of its drawback is that in some configurations, the number of points around the neck is weak, so as the parabolas that fit the strain at necking can be tricky to determine.

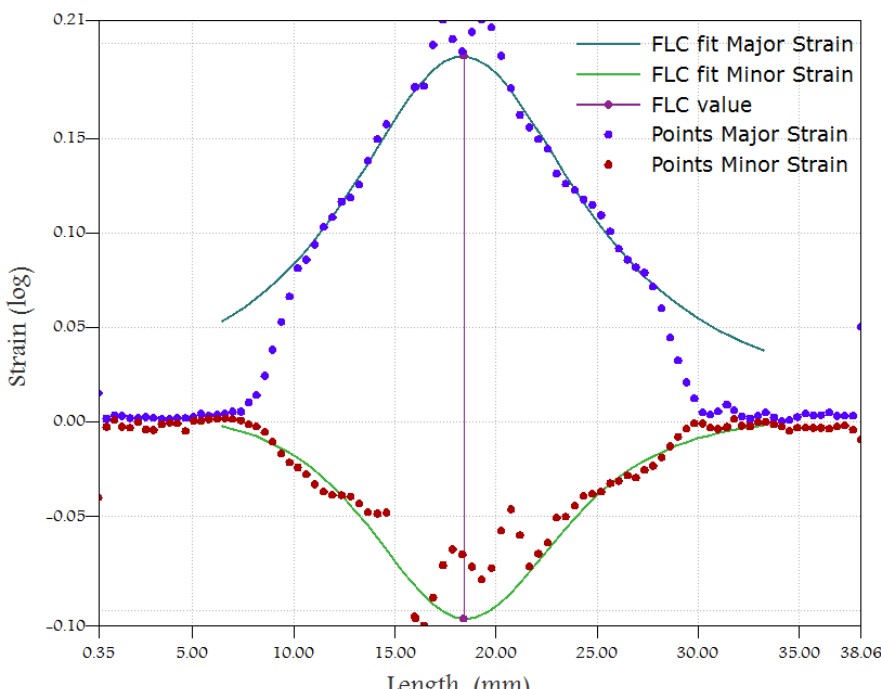

**Figure 7.** ISO 12004-2 method: major and minor strains as a function of the length for a selected section just before fracture (points) and their fitting with an inverse parabola (full lines). The vertical line gives the major and minor strains of the FLC for this strain path.

#### 2.3.2. Time Dependent Method

An alternative to the previous method consists in using time-dependent analysis, based on the follow-up of the strain acceleration in the neck [7]. In this method, a regression is used to determine the change from elastic behavior to plastic behavior, based on an analysis of the progression of the time derivative of the strain rate to determine the beginning of necking, which corresponds to the maximum of the correlation coefficient. Due to the noise in the full field measurement, it can be difficult to obtain a strain acceleration evolution that is easy to exploit. For this reason the data curve is firstly smoothed

by a local regression using the weighted linear least squares method using a second degree polynomial model (Figure 8).

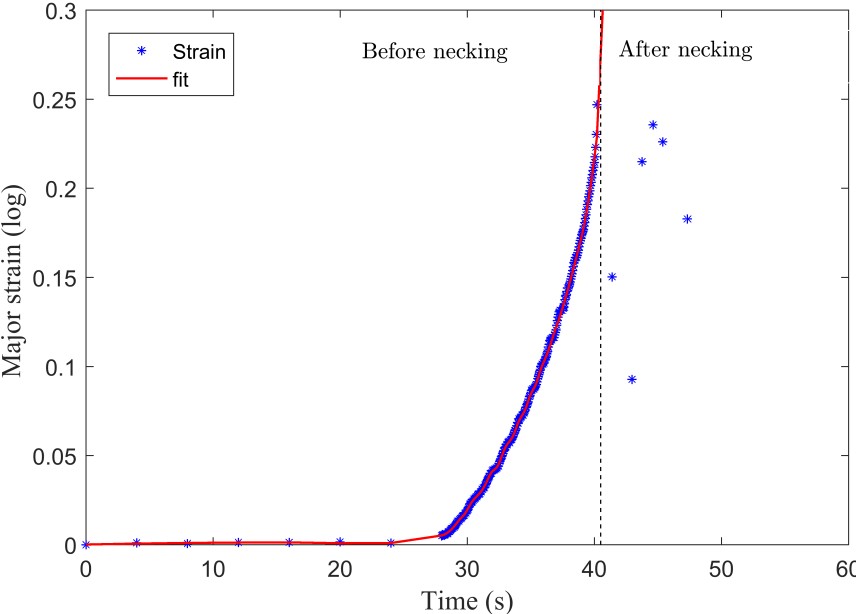

**Figure 8.** Determination of onset of localized necking for the circular sample (65 mm) of pure Cu: Evolution of the major strain as a function of time.

The evolution of the strain rate presented Figure 9 is obtained by the numerical derivative of the previous curve. Despite the smoothing treatment, it is difficult to obtain a perfectly stable signal. Indeed, a linear evolution can be identified at the beginning of the test, then the derivative of the strain rate with respect to the time increases (for $t = 43$ s) to reach a maximum with the beginning of the localization of necking.

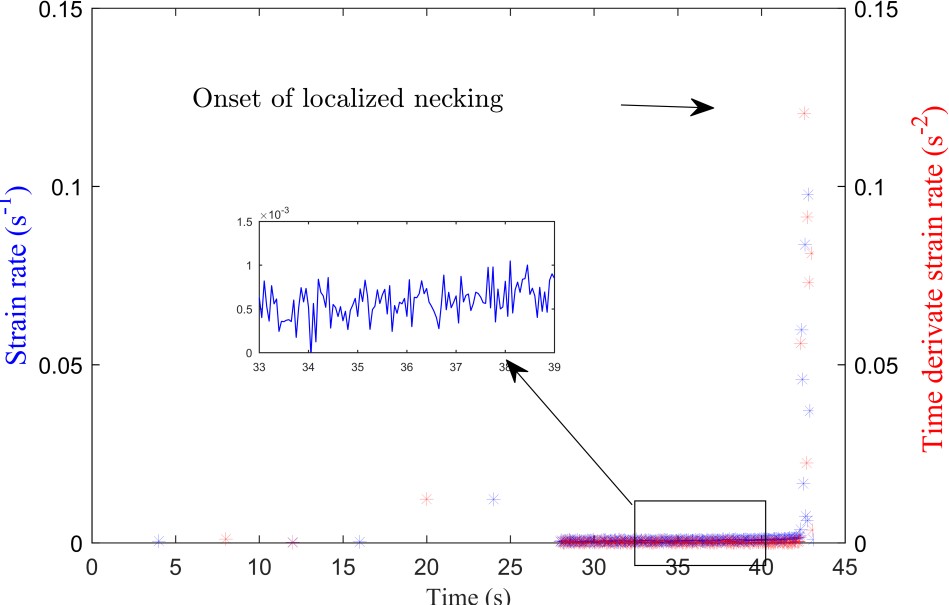

**Figure 9.** Determination of the onset of necking using the time-dependent method for the circular sample of pure Cu (65 mm): evolution of major strain rate (blue curve), major principal acceleration history (red curve).

The principle of the time-dependent method is therefore based on the fact that the strain rate increases highly due to the necking of the sample [37]. The maximum of the curve corresponds to the instant of the onset of necking. The corresponding major and minor strain values represent the point of the FLC for this strain path.

Forming limit points obtained with both methods, i.e. ISO-12004 and time-dependent methods, are shown in Figure 10 for both materials, for all the tests that were available in order to check the reproducibility of the results. A slight data dispersion is observed for the time-dependent method, and as reported in previous studies [37,38], FLCs obtained using the time-dependent method are often higher than the ISO-12004 method, confirming that FLCs are strongly dependent on the determination method. Fits of data points with linear regressions in drawing and stretching areas are also plotted in Figure 10.

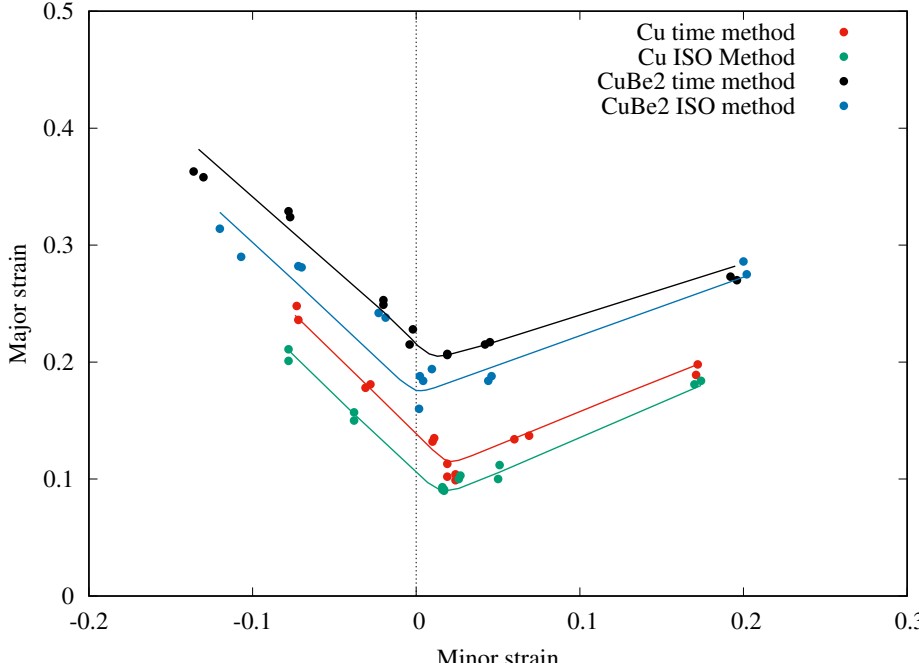

**Figure 10.** Forming limit curves of pure Cu and CuBe2 alloy: comparison of ISO-12004 and time-dependent methods.

Figure 10 shows that the time-dependent method may be prefered to determine the forming limits since it has the advantage of being a local method, independent of many geometrical parameters involved in the ISO method (size of the window used to fit the inverse parabola for example). Thus, it can be directly applied to problems with non-negligible strain gradients as previously mentioned in [39]. Furthermore, [40] have shown the effectiveness of this methodology for determining the limits of formability in the case of thin-walled aluminum tubes. It should also be observed that the time-dependent average fits consist merely for both materials in a shift of the ISO-12004 curves both in the range of higher major and minor strains. This shift in the direction of the minor strains moves artificially the curve from the minimum for a plane strain path from the null minor strain, which justifies taking into account the effects depending on the process in the next section.

## 3. Compensation of the FLC for Process-Dependent Effects

Many parameters can affect the forming limits that can be reached by a material when it is plastically deformed. Among them, the curvature of the sheet, non-linear strain paths and the contact pressure appear to be the main ones [24]. These parameters affect the limits of necking and may contribute to overestimating them. In the following three parts, a procedure to compensate

the strain measurements at the localized onset of necking from the Nakazima tests are performed. This method proposed by Min et al. [24] takes into account the effects of curvature, the non-linear strain paths, and the influence of the non-zero pressure stresses due to the contact between the tool and the specimen. The purpose of this compensation is to eliminate their combined effects that occur during the experimental tests and to obtain a FLC closer to the Marciniak method [41], considered as the reference method. It has to noticed that such careful treatments are very few in the literature.

### 3.1. Determination of the Strain Path in the Thickness of the Sheet

The effect of curvature on strain and its impact on the determination of localized necking conditions has been noticed since the first use of FLC to evaluate the formability of a given material [41]. In order to determine the conditions responsible for a localized instability in a deformed sheet, it is necessary to evaluate the severity of the forming conditions on each layer due to the curvature of the punch. Let us consider a set of major and minor strain points $(\epsilon_{1,j}^X, \epsilon_{2,j}^X)$ with $j = (0, N)$, where $N$ is the number of images given by the DIC measurement system and X = (O,M,I) the outer, middle and inner layers of the sheet, respectively, as shown in Figure 11.

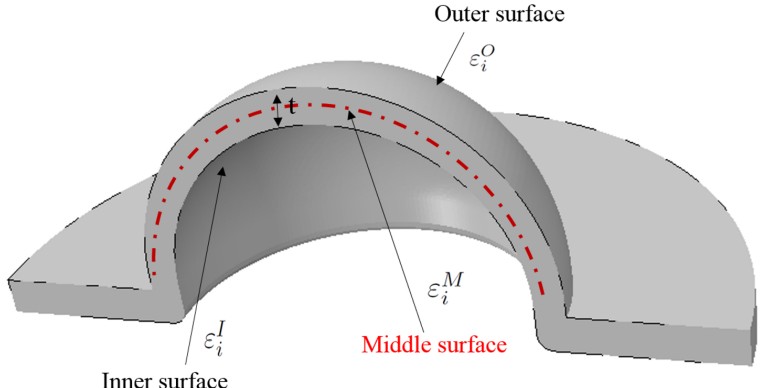

**Figure 11.** Definition of strains in the layers of the sheet: Outer surface ($\epsilon_i^O$), Inner surface ($\epsilon_i^I$) and Middle layer ($\epsilon_i^M$); $i = 1, 2$ are the principal directions.

The strain paths for the middle and inner surfaces are not accessible by the DIC measurement system, but they can be determined as a function of the strain of the outer surface, the main curvatures and the thickness of the sheet. Firstly, to determine the strain on non-accessible surfaces, it is necessary to determine the curvatures of the sheet on the outer surface $\kappa_{i,j}^O = 1/R_{i,j}^O$. For this, the radii of curvature $(R_{1,j}^X, R_{2,j}^X)$, are determined at each instant by adjusting the curvature of the sample to a sphere using the Aramis software according to Figure 12.

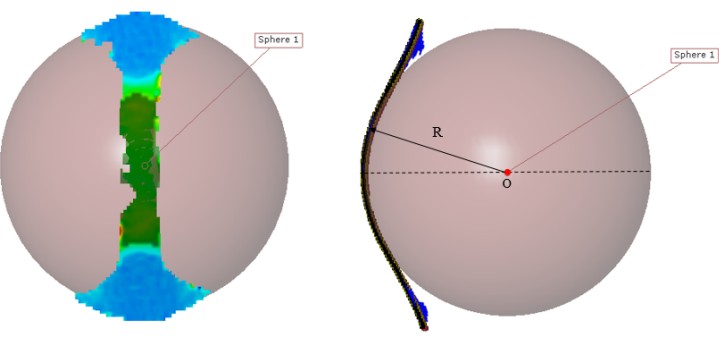

**Figure 12.** Aramis procedure for the determination of the radius of curvature, adjusting the curvature of the sample to a sphere.

Due to the thinning of the sheet occurring during the deformation process, it is necessary to determine the local thickness of the sheet for a non-flat surface, taking into account the curvature and the strain measurements of the outer surface [10]. Let us denote by $t_0$ the initial thickness, the variation of thickness $t_j$ of the sheet for times $j = (1, N)$ may be evaluated accurately by the following cubic Equation (1):

$$at_j^3 + bt_j^2 - ct_j - d = 0 \tag{1}$$

where $a, b, c, d$ are functions of the principal strains $\epsilon_{1,j}^O$ and $\epsilon_{2,j}^O$ on the outer surface, the curvature radii $R_{1,j}^O$ and $R_{2,j}^O$ and $\epsilon_{V,j}^e$ the elastic volume change, which is neglected in a first approach in the calculation of the thickness, since the elastic dilatancy is weak in metallic materials [24]. After determining the thickness by Equation (1) and curvatures at each instant, the principal strains on the middle and inner surfaces are respectively defined by:

$$\epsilon_{i,j}^M = \epsilon_{1,j}^O + \ln(1 - t_j \frac{\kappa_{i,j}^O}{2}) \text{ for } i = 1, 2 \text{ and } j = (1, N) \tag{2}$$

$$\epsilon_{i,j}^I = \epsilon_{1,j}^O + \ln(1 - t_j \kappa_{i,j}^O) \text{ for } i = 1, 2 \text{ and } j = (1, N) \tag{3}$$

Figure 13 shows the strain paths calculated on the middle and inner surfaces up to the onset of localized necking from the strain path measured on the outer surface for a 45 mm width specimen of pure Cu. Such tendency is representative of all other samples for both materials. It is observed that as ultra-thin sheets are considered, the difference between the strains of the different layers is weak and the effect of curvature could be neglected for such thickness and punch diameter. Consequently, no influence was observed on the FLC due to curvature effects.

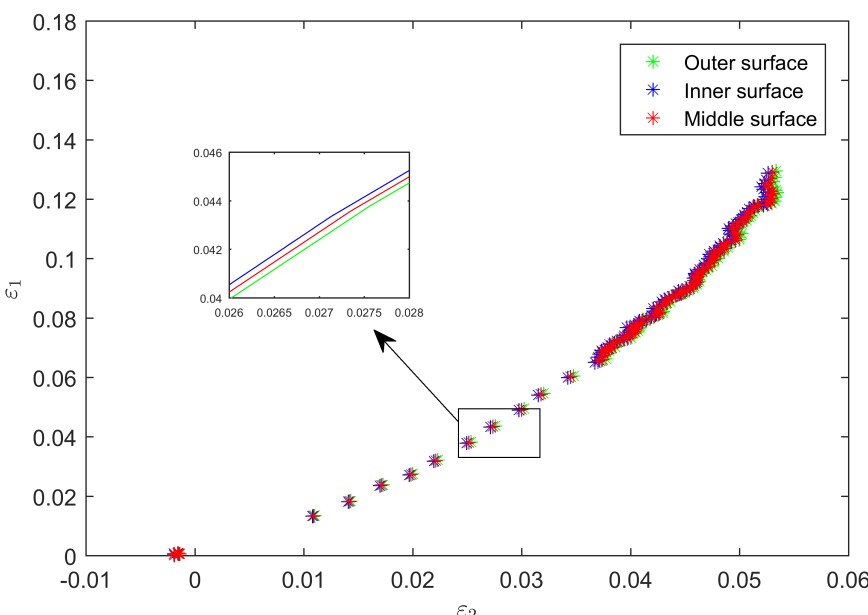

**Figure 13.** Strain paths on the outer, middle and inner surfaces of a 45 mm width specimen of pure Cu.

### 3.2. Effects of Non-Linearity of the Strain Paths

Conversely to the weak influence of curvature effect, it was observed in Figures 5 and 6 that due to the geometry of the punch, most of the strain paths are non-linear for both materials. According to [24], non-linear strain paths affect the forming limits. In this section, a correction method for a constitutive model involving normal anisotropy is retained, which properties along the thickness are different from those in the sheet plane. The objective of this compensation is to obtain a FLC that applies under in-plane plane-stress deformation restricted to perfectly linear strain paths. The behaviour is

considered as isotropic in the sheet plane, according to the values of $\bar{r}$ close to 1, particularly for CuBe2 alloy. As in the previous part, it is considered that the elastic strain is negligible [41]. For this, it is necessary to ensure that the deformation increments between two successive instants are sufficiently large to neglect the elastic contribution, which amounts to considering that the total deformation increments are only plastic. Since the variation of the strains between two successive images is often very small, it was necessary to delete data points from the set of data, keeping only $\hat{N}$ increments with $\hat{N} < N$ so that the following condition is satisfied:

$$\sqrt{\left(\Delta\epsilon_{1,j}^X - \Delta\epsilon_{1,j-1}^X\right)^2 + \left(\Delta\epsilon_{2,j}^X - \Delta\epsilon_{2,j-1}^X\right)^2} \geq \Delta\epsilon_{\min} \text{ for } j = (2, \hat{N}) \tag{4}$$

and:

$$\sqrt{\left(\Delta\epsilon_{1,j}^X\right)^2 + \left(\Delta\epsilon_{2,j}^X\right)^2} \geq \Delta\epsilon_{\min} \text{ for } j = 1 \tag{5}$$

This processing is applied step by step, comparing each pair of data to the previous data pair until the pair of data from which the previous relationships are verified is obtained. $\Delta\epsilon_{\min}$ is set to 0.01, but the impact of increasing or decreasing this tolerance by a factor of two is weak [24]. After this treatment, total strain increments can be considered as plastic:

$$\Delta p_{i,j}^X \simeq \Delta\epsilon_{i,j}^X \text{ for } j = (1, \hat{N}) \tag{6}$$

The next step is to remove the effect of non-linear strain paths on the determination of the limit condition on each surface. As described in [24], it consists in integrating the increments of effective plastic strain for each strain path and calculating the corresponding stress limit $(\sigma_1, \sigma_2)$, since the stress-based necking limits are insensitive to non-linear strain paths as mentioned in [42]. Although it should be noted that the Nakazima test does not provide measurement of the stress condition, Min et al. [24] described the following procedure to use the strain path data to calculate the stress conditions for each strain path. This correction method is based on a pressure-insensitive quadratic yield function, given by the following Equation (7) involving the normal anisotropy coefficient $\bar{r}$:

$$\bar{\sigma}(\sigma_1, \sigma_2) = \sqrt{\sigma_1^2 + \sigma_2^2 - \frac{2\bar{r}}{1 + \bar{r}}\sigma_1\sigma_2} \tag{7}$$

where $\sigma_1$ and $\sigma_2$ are principal stresses. This assumption is chosen because the previous relation between the stress and the strain can be easily reversed. Considering a more complex constitutive law requires using a User subroutine (Umat) to solve the equation incrementally. This was not done in this study. The main stresses at the beginning of the localized necking along the plastic deformation path are given by:

$$\begin{pmatrix} \sigma_1^X \\ \sigma_2^X \end{pmatrix} = \begin{pmatrix} 1 \\ \alpha^X \end{pmatrix} \frac{\sigma_y(\bar{\epsilon}_p^X)}{\bar{\sigma}_y(1, \alpha^X)} \tag{8}$$

with $\sigma_y(\bar{\epsilon}_p^X)$ is the hardening law. In our case, the Hollomon's equation is chosen, which is given by:

$$\sigma_y(\bar{\epsilon}_p) = K\bar{\epsilon}_p^n \tag{9}$$

where $\sigma_y$ is the yield stress in tension, $\bar{\epsilon}_p$ is the equivalent plastic strain, $K$ and $n$ are material constants obtained by identifying experimental data from a tensile test in the rolling direction (see Table 2 and Figure 3). $\bar{\sigma}_y(1, \alpha^X)$ is a two-parameter function of the form given by Equation (7) while replacing $(\sigma_1, \sigma_2)$ with variables $(1, \alpha^X)$ where $\alpha^X$ is the ratio of main stresses on the three layers at the beginning of the localized necking, given by:

$$\alpha^X = \frac{(1+\bar{r})\Delta p^X_{2,\hat{N}} + \bar{r}\Delta p^X_{1,\hat{N}}}{(1+\bar{r})\Delta p^X_{1,\hat{N}} + \bar{r}\Delta p^X_{2,\hat{N}}} \tag{10}$$

where $(\Delta p^X_{1,\hat{N}}, \Delta p^X_{2,\hat{N}})$ are the last increase in plastic deformation before the localized necking starts. The equations defining the strain limits for a perfectly linear strain path compatible with the stress limits defined by Equation (8) are given as follows:

$$\begin{pmatrix} p^X_1 \\ p^X_2 \end{pmatrix} = \begin{pmatrix} \Delta p^X_{1,\hat{N}} \\ \Delta p^X_{2,\hat{N}} \end{pmatrix} \frac{\bar{\epsilon}^X_p}{\dot{\bar{\epsilon}}_p(\Delta p^X_{1,\hat{N}}, \Delta p^X_{2,\hat{N}})} \tag{11}$$

where $\dot{\bar{\epsilon}}_p(\Delta p^X_{1,\hat{N}}, \Delta p^X_{2,\hat{N}})$ is a function defined by Equation (12) evaluated at the last increment of the strain leading to the appearance of localized necking $(\Delta p^X_{1,\hat{N}}, \Delta p^X_{2,\hat{N}})$.

$$\dot{\epsilon}_p(\dot{p}_1, \dot{p}_2) = \frac{1+\bar{r}}{\sqrt{1+2\bar{r}}}\sqrt{\dot{p}_1^2 + \dot{p}_2^2 + \frac{2\bar{r}}{1+\bar{r}}\dot{p}_1\dot{p}_2} \tag{12}$$

While $\bar{\epsilon}^X_p$ is the solution of the following Equation (13):

$$\bar{\epsilon}^X_p = \sum_{j=1}^{j=\hat{N}} \dot{\bar{\epsilon}}_p(\Delta p^X_{1,j}, \Delta p^X_{2,j}) \tag{13}$$

The results of the strain calculations using Equation (11) are compared in Figure 14 with the measured strain (without correction) at the onset of localized necking using the time dependent method.

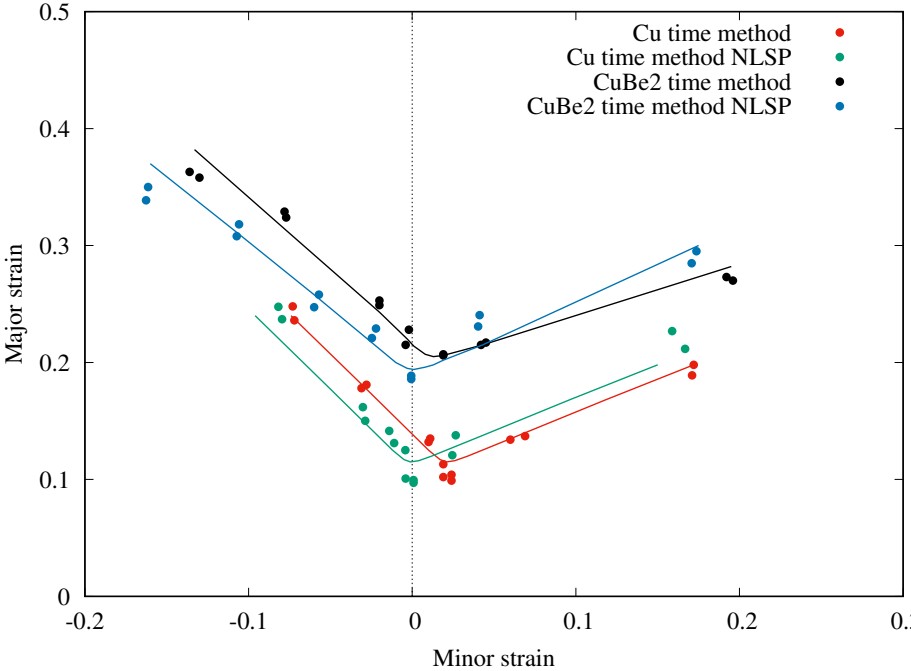

**Figure 14.** Process corrections for non-linear strain path effects (NLSP) in the Nakazima tests for the pure Cu and the CuBe2 alloy: strain limits are obtained with the time-dependent method.

The effect of the non-linear strain path correction applies differently for both materials. For the pure Cu, it consists merely in a shift of the FLC to lower minor strains, so as the minimum of the curve matches the minimum in plane strain. For the CuBe2 alloy, it moves the minimum of the strain forming limit curve for these tests from a low positive value to a minor strain of substantially zero corresponding to the theoretical minimum in plane strain. Thus, it reduces the necking strain in the

vertical direction by approximately 3.5 to 4%. Moreover, it increases the expansion necking strain by approximately 1 to 1.5% in the vertical direction. The non-linear strain path corrections were more pronounced on the drawing side of the diagram. These findings are consistent with those of Min et al. [41] for which differences in necking limits obtained by Nakazima testing are mainly due to friction conditions that are different for both materials, and that can be eliminated and fully explained by application of the compensation method.

*3.3. Pressure Effects*

In this section, a last correction step is applied to determine the forming limits as a function of the strain while taking into account the effect of the pressure due to the contact between the sheet and the punch. Indeed, there is no correction of the strain calculated on the outer surface because the pressure is zero on this surface. However, the pressure becomes significant on the middle and inner layers. Let us denote the contact pressure $p = -\sigma_3$ where $\sigma_3$ is the stress in the thickness direction on the inner surface, the compensated critical stresses for the three layers are respectively given by:

$$\begin{pmatrix} \sigma_1^O \\ \sigma_2^O \end{pmatrix} = \begin{pmatrix} 1 \\ \alpha^O \end{pmatrix} \frac{\sigma_y(\bar{\epsilon}_p^O)}{\bar{\sigma}_y(1, \alpha^O)} \tag{14}$$

$$\begin{pmatrix} \sigma_1^M \\ \sigma_2^M \end{pmatrix} = \begin{pmatrix} 1 \\ \alpha^M \end{pmatrix} \frac{\sigma_y(\bar{\epsilon}_p^M)}{\bar{\sigma}_y(1, \alpha^M)} - \frac{p}{2} \begin{pmatrix} 1 \\ 1 \end{pmatrix} \tag{15}$$

$$\begin{pmatrix} \sigma_1^I \\ \sigma_2^I \end{pmatrix} = \begin{pmatrix} 1 \\ \alpha^I \end{pmatrix} \frac{\sigma_y(\bar{\epsilon}_p^I)}{\bar{\sigma}_y(1, \alpha^I)} - p \begin{pmatrix} 1 \\ 1 \end{pmatrix} \tag{16}$$

where $p$ is calculated according to the thickness $t$ of the sheet and the curvature $\kappa^X$ by the following equation (Equation (17)):

$$p = t\kappa^I(1 + \frac{1}{2}t\kappa^I)(\sigma_1 + \sigma_2) \quad \text{where} \quad \kappa^I = \frac{\kappa^O}{1 - t\kappa^O} \tag{17}$$

From the stress limits defined by main stresses $(\sigma_1, \sigma_2)$ the corresponding point on the strain limits for a linear strain path $(p_1, p_2)$ may be obtained from:

$$\begin{pmatrix} p_1 \\ p_2 \end{pmatrix} = \begin{pmatrix} 1 \\ \beta \end{pmatrix} \frac{\bar{\epsilon}_p^X}{\dot{\bar{\epsilon}}_p(1, \beta)} \tag{18}$$

where $\beta$ can be calculated as:

$$\beta = \frac{(1 + \bar{r})\alpha + \bar{r}}{1 + \bar{r} + \alpha\bar{r}} \tag{19}$$

$\dot{\bar{\epsilon}}_p(1, \beta)$ is a two-parameter function of the form given by Equation (7) while replacing $(\sigma_1, \sigma_2)$, the variables $(1, \beta)$ and $\bar{\epsilon}_p^X$ from the hardening model according to:

$$\bar{\sigma}_y(\sigma_1, \sigma_2) = \sigma_y(\bar{\epsilon}_p) \tag{20}$$

The limit strains, corrected for the non-linear strain path effects, are then corrected for the through-thickness contact stress using Equations (14)–(17). Figure 15 displays that the pressure correction causes a moderate variation of the data points that do not affect the global tendency given by the linear regressions. The obtained results are in good agreement with those reported in [24,33] in the case of the AA5182-O alloy and an advanced high strength steel.

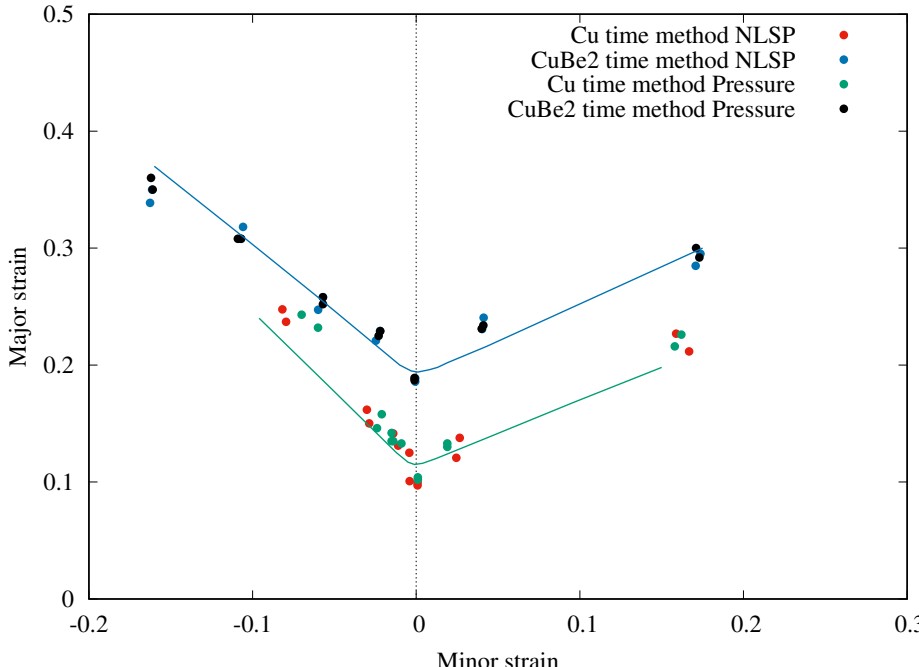

**Figure 15.** Comparison of forming limits after correction for non-linear strain paths and contact pressure effects from the Nakazima tests of pure Cu and CuBe2 alloy.

Finally, the correction method adopted in this work to obtain linear deformation paths up to failure is relevant for ultra-thin sheets, since the strain paths obtained in the Nakazima tests exhibited non-linear strain paths. This could be confirmed by microforming Marciniak tests [35], but these tests are generally performed on a much lower scale than this one (punch of 1.5 mm compared to 35 mm), which is likely to exacerbate other local and scale effects. The corrected FLCs for the pure Cu and CuBe2 alloy in Figure 15 are in good agreement with the results presented by Noder et al. [33] for the study of the formability of an AA5182 alloy 1.55 mm thick and DP980, with a sheet thickness of 1.2 mm. These resulting FLCs may be considered as the true forming limits of the selected materials, for perfectly linear strain paths in the absence of normal pressure through the thickness.

## 4. Conclusions

This work aims to characterize experimentally the forming limits of ultra-thin sheets (thickness 0.1 mm) of copper materials that are used for connection applications. Nakazima tests are performed with a specific device in order to determine experimentally the onset of rupture and the forming limit curves of both materials. Two methods were used to determine necking, the time-dependent method and the position-dependent method (ISO-12004 method). As most of the tests present non-linear strain paths, a compensation method is used to correct the measurement of FLCs for non-linear strain paths, curvature effects, and pressure across thickness. Therefore, FLCs of ultra-thin sheets of pure copper and copper beryllium alloy, independent of the process and with a linear strain path hypothesis have been carefully established. The main findings are summarized as follows:

1. Our Nakazima test device is relevant for the study of the formability of ultra-thin sheets. Compared to other Nakazima or Marciniak devices that have been developed specifically for ultra-thin sheets, it remains macroscopic, avoiding any local or scale effects that can arise with very small tools.
2. The ISO 12004-2 method underestimates the FLCs for both materials. The time-dependent method, with its numerical based determination of the onset of necking, offers the possibility of

generating objective and reproducible FLCs without any influence of the user. This method is more stable and enhance the measurement of the formability.

3. Concerning the compensation method, the difference between the strains of the different layers is weak and the effect of curvature can be neglected for such thickness and punch diameter. Consequently, no influence was observed on the FLCs due to curvature effects.

4. The correction process leads to a reduction of the necking strain in the major strain direction for both materials, but an increase in the expansion necking strain. The instantaneous strain path at the limit strain converges to the plane strain and the non-linear strain path correction significantly reduced the minor strain.

**Author Contributions:** Investigation, data curation and writing—original draft preparation, N.A.; investigation and methodology, C.H.P.; writing—review and editing, N.G.; supervision, writing—review and editing, P.-Y.M. All authors have read and agreed to the published version of the manuscript.

**Funding:** This research received no external funding.

**Conflicts of Interest:** The authors declare no conflict of interest.

**Data Availability:** The raw/processed data required to reproduce these findings cannot be shared at this time as the data also forms part of an ongoing study.

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
