# Peer review of "Development of a Nakazima Test Suitable for Determining the Formability of Ultra-Thin Copper Sheets"

_metals, doi:10.3390/met10091163_

Round 1
Reviewer 1 Report
The authors present an interesting study concerning an accurate method for determining FLC´s for ultra-thin sheet metals from both, copper and CuBe alloy. The authors used Nakazima test which was adapted (new tools) to the very low thickness of the tested sheets. Two different methods are used to estimate necking. First, they used the method proposed by the ISO 12004-2 and the second one time-dependent method. The authors applied the compensation method, which combines the influence of nonlinear strain paths, the effect of curvature and pressure. A specific procedure was developed to correct the FLCs for tested materials.
Although the use of time-dependent methods has been previously proposed (see comments 1), the authors wanted to introduce a correction routine to generating objective and reproducible FLCs. This can be stated as the novelty of the work a detailed analysis of the influence of the effects mentioned above on the position of the FLCs.
In brief, the work has potential but needs to be improved. I add these comments because I think that are fundamental for a proper presentation of the manuscript. Moreover, in my opinion, the quality of the text could also be improved.
COMMENT 1: In lines 189-220
According to Volk, Hora, Hotz, Merklein, Slota, and others, basically, four different time-dependent methods are commonly used:
- Correlation coefficient method
- Linear best fit method
- Gliding correlation coefficient
- Gliding difference of mean to median.
Which method was used by the authors? In my opinion, this part of the manuscript should be improved.
Literature (for example):
- https://link.springer.com/article/10.1007/s12289-010-1012-9
- https://www.scientific.net/KEM.549.397
- W. Hotz, J. Timm: Experimental detection of forming limit curves (FLC), In: Numisheet 2008, Interlaken/Switzerland, Ed. P. Hora, 2008, p. 271-278
- M. Merklein, A. Kuppert, S. Mütze, A. Geffert: New Time Dependent Method for Determination of Forming Limit Curves Applied to SZBS800. In: IDDRG 2010, Graz, Verlag der Technischen Universität Graz, Vol. 50, 2010, p. 489-498
- https://www.qip-journal.eu/index.php/ams/article/view/639
COMMENT 2: In lines 192-193, the authors state: „For this reason the data curve is firstly smoothed using an exponential function.“
Why did the authors use the exponential function? What is the value of the coefficient of determination (R2)?
COMMENT 3: In line 195, the authors state: „...it is difficult to obtain a perfectly stable signal“.
I think that using a suitable filter in Matlab would be possible.
COMMENT 4: In Figure 9: I suggest adding units on the axis.
SUGGESTION 1: In line 13, replace „non linear” by “non-linear”
SUGGESTION 2: In Table 2, replace „paramters” by “parameters”
Author Response
Reviewer #1
-
Comment 1: In lines 189-220, according to Volk, Hora, Hotz, Merklein, Slota, and others, basically, four different time-dependent methods are commonly used: Correlation coeffi- cient method, Linear best fit method, Gliding correlation coefficient, Gliding difference of mean to median. Which method was used by the authors? In my opinion, this part of the manuscript should be improved.
According to this remark, this point has been clarified in the revised paper:
An alternative to the previous method consists in using time-dependent analysis, based on the follow-up of the strain acceleration in the neck [7]. In this method, a regression is used to determine the change from elastic behavior to plastic behavior, based on an analysis of the progression of the time derivative of the strain rate to determine the beginning of necking, which corresponds to the maximum of the correlation coefficient. Due to the noise in the full field measurement, it can be difficult to obtain a strain acceleration evolution that is easy to exploit. For this reason the data curve is firstly smoothed by a local regression using the weighted linear least using a second degree polynomial model (Fig.8).
-
Comment 2: In lines 192-193, the authors state: "For this reason the data curve is firstly smoothed using an exponential function." Why did the authors use the exponential func- tion? What is the value of the coefficient of determination (R2)?
This was a mistake. The authors have corrected this statement and have clarified this point according to comment 1.
-
Comment 3: In line 195, the authors state: "...it is difficult to obtain a perfectly stable signal". I think that using a suitable filter in Matlab would be possible.
We used a Matlab filter (rloess) which is based on a local regression using weighted linear least squares and a second degree polynomial model. That assigns lower weight to outliers in the regression.
-
Comment 4: In Figure 9: I suggest adding units on the axis. Done.
-
Suggestion 1: In line 13, replace "non linear” by “non-linear” . Done.
-
Suggestion 2: In Table 2, replace "paramters” by “parameters”. This mistake has been corrected.

Reviewer 2 Report
The topic is interesting and the manuscript is well structured. Minor revision is suggested.
As emphasized by the authors, the purpose of the compensation is to eliminate the combined effects of the curvature of the sheet, nonlinear strain paths and the contact pressure, so as to obtain a FLC closer to the Marciniak method. However no related results have been provided to support this conclusion.
Author Response
Reviewer #2
1. As emphasized by the authors, the purpose of the compensation is to eliminate the combined effects of the curvature of the sheet, nonlinear strain paths and the contact pressure, so as to obtain a FLC closer to the Marciniak method. However no related results have been provided to support this conclusion.
In their paper, Min et al. [24] have shown that this compensation method is able to transform the FLC obtained from Nakazima tests so as to be superimposed with those obtained using Marciniak test. In our study, it was not possible to perform Marciniak tests with ultra-thin sheets (we don’t have the device). So, according to [24], we have supposed that using this method was able to transform our FLCs (starting from Fig.10 to Fig.15) in order to obtain the minimum in major strain for a plane strain mode, i.e. for a null minor strain. This is consistent with our results and this is discussed just before the conclusion:
Finally, the correction method adopted in this work to obtain linear deformation paths up to failure is relevant for ultra-thin sheets, since the strain paths obtained in the Nakazima tests exhibited non-linear strain paths. This could be confirmed by microforming tests [35], but these tests are generally performed on a much lower scale than this one of 1.5 mm compared to 35 mm), which is likely to exacerbate other local and scale effects. The corrected FLCs for the pure Cu and CuBe2 alloy in Fig.15 are in good agreement with the results presented by Noder et al [33] for the study of the formability of an AA5182 alloy 1.55 mm thick and DP980, with a sheet thickness of 1.2 mm. These resulting FLCs may be considered as the true forming limits of the selected materials, for perfectly linear strain paths in the absence of normal pressure through the thickness.

Reviewer 3 Report
This paper developed a Nakazima test suitable for determining the formability of ultra-thin copper sheets. In the review, I thought the paper is not suitable for publishing in "Metal" journal. The comments are as following.
- Could the Nakazima test apply to the other metals? The author should give more references.
- The author should give more detailed discussions on the difference between traditional test and Nakazima test.
- The abstract should be modified to summarize the result of this study.
Author Response
Reviewer #3
1. Could the Nakazima test apply to the other metals? The author should give more refer- ences. The author should give more detailed discussions on the difference between traditional test and Nakazima test.
Ok. The authors have added a new reference (Grolleau et al.) that compares classical tests together with both Nakazima and Marciniak tests on several materials (an aluminium alloy and two steels) to determine forming limit curves. The authors have taken into account these remarks in the revised Introduction:
The experiments carried out by Grolleau et al [23] using V-bending, notched tension and Nakazima specimens on 2024-T351 aluminum specimens, DP450 steel and DP980 steel show that for these materials, the Nakazima test provides a reliable estimation of the strain at fracture for the plane stress state. None of the two other tests (V-bending and notched tension) provides reliable results for the three materials. As a result, only with the help of the Marciniak and Nakazima tests, the forming limit curves can be fully drawn, despite that in the case of the Nakazima test there are sometimes difficulties related to the measurement of the deformations. However, the curvature of the blanks after forming, due to the hemispherical shape of the punch, and the friction between the blank and the tools can skew the deformation measurements in the critical zone [24].
2. The abstract should be modified to summarize the result of this study.
Ok. A sentence summarizing the results has been added at the end of the abstract:
The curvature effect for such thickness and punch diameter on the FLCs is weak. Moreover, the instantaneous strain path at the limit strain converges to the plane strain state.

Round 2
Reviewer 3 Report
I think the author mistake my meaning. I would the author to summarize the abstract. It means that the abstract is too long and it needs to shorten.
Author Response
Ok. This has been done. Here is the new abstract:
The objective is to propose an accurate method for determining the forming limit curves (FLC) for ultra-thin metal sheets which are complex to obtain with conventional techniques. Nakazima tests are carried out to generate the FLCs of a pure copper and a copper beryllium alloy with a thickness of 0.1 mm. Because of the very small thickness of the sheets, the standard devices and the know-how of this test are no longer valid. Consequently, new tools have been designed in order to limit friction effect. Two different methods are used and compared to estimate the necking: the position-dependent measurement method (ISO Standard 12004-2), and the time-dependent method based on the analysis of the derivatives of the planar strain field. It is shown that the ISO standard method underestimates the forming limit curves. As the results present non linear strain paths, a compensation method is applied to correct the FLCs for the tested materials, which combines the effects of curvature, nonlinear strain paths and pressure. The curvature effect for such thickness and punch diameter on the FLCs is weak. The results show that this procedure enables to obtain FLCs that are close to those determined by the reference Marciniak method, leading to a minimum in major strain that converges to the plane strain state.
